# Analysis of the Effect of Fiber Orientation on Mechanical and Elastic Characteristics at Axial Stresses of GFRP Used in Wind Turbine Blades

**DOI:** 10.3390/polym15040861

**Published:** 2023-02-09

**Authors:** Ciprian Ionuț Morăraș, Viorel Goanță, Dorin Husaru, Bogdan Istrate, Paul Doru Bârsănescu, Corneliu Munteanu

**Affiliations:** 1Mechanical Engineering, Mechatronics and Robotics Department, Mechanical Engineering Faculty, “Gheorghe Asachi” Technical University of Iasi, 700050 Iasi, Romania; 2Fluid Mechanics, Fluid Machines and Fluid Power Systems Department, Machine Manufacturing and Industrial Management Faculty, “Gheorghe Asachi” Technical University of Iasi, 700050 Iasi, Romania; 3Technical Sciences Academy of Romania, 26 Dacia Blvd., 030167 Bucharest, Romania

**Keywords:** fibre orientation, wind turbine blade, tensile, strain gauge, GFRP, SEM, FEM

## Abstract

Due to its physical and mechanical properties, glass-fiber-reinforced polymer (GFRP) is utilized in wind turbine blades. The loads given to the blades of wind turbines, particularly those operating offshore, are relatively significant. In addition to the typical static stresses, there are also large dynamic stresses, which are mostly induced by wind-direction changes. When the maximum stresses resulting from fatigue loading change direction, the reinforcing directions of the material used to manufacture the wind turbine blades must also be considered. In this study, sandwich-reinforced GFRP materials were subjected to tensile testing in three directions. The parameters of the stress–strain curve were identified and identified based on the three orientations in which samples were cut from the original plate. Strain gauge sensors were utilized to establish the three-dimensional elasticity of a material. After a fracture was created by tensile stress, SEM images were taken to highlight the fracture’s characteristics. Using finite element analyses, the stress–strain directions were determined. In accordance to the three orientations and the various reinforcements used, it was established that the wind turbine blades are operational.

## 1. Introduction

With the start and development of the industrialization process, energy consumption, both globally and locally, has grown rapidly, but it has also contributed to an increase in greenhouse gas emissions, which have dramatic consequences for global warming. The development of renewable energy will make a considerable contribution to mitigating climate change. In recent years, wind energy has been an important source of energy, and it is still growing [1]. In the last decade, GFRP reinforcement and other composite materials have gained unprecedented popularity due to their ability to resist corrosion and solve long-standing structural problems.

The main way to reduce the price of electricity supplied by wind turbines is to increase the installed power of the turbines, which leads to an increase in the rotor diameter and thus the blade length [2]. Currently the largest wind turbine has an installed power of 15 MW and a rotor diameter of 236 m [3]. Such rotor diameters could not have been achieved without the use of composite materials in blade construction. In the rotor assembly of a wind turbine blade (WTB), the critical component is the blade. A statistical study of wind turbine failures shows that more than 65% of failures are due to blade breakage [4]. For this reason, it is necessary to study in depth the mechanical and elastic characteristics of the composite material used to make the blades and to develop blade strength calculations.

The most widely used material for blade manufacturing is GFRP, due to its mechanical and elastic characteristics and environmental resistance, as well as its lower price compared to other materials [5,6,7].

The main function of reinforcement in fiber-reinforced composites is to take up the load along the length of the fiber and provide strength and stiffness in one direction. Several types of fibers can be used for reinforcement, differing in stiffness, elastic characteristics and tensile strength. Glass fibers are considered the predominant reinforcement for polymer matrix composites due to their high electrical insulation properties, low moisture sensitivity and high mechanical properties. Glass fibers have equal or better characteristics than steel under certain reinforcement and load conditions. Elements that influence the mechanical properties of GFRP composites include fiber type, matrix, relative amounts of constituents and fiber orientation [8]. The quality of the interface between matrix and fibers also has a major impact on the mechanical properties of GFRP. In addition, the post polymerization and polymerization system can have an impact on the final properties of the composite materials. Ultimate tensile strength, stiffness and creep and uniaxial tensile comportment are some of the mechanical properties of GFRP materials to be considered and accurately determined [9]. The mechanical properties of GFRP composites render these materials ideal for building and strengthening structures.

Steel and other metals have isotropic properties, which means they offer equal strength, deformation and elastic characteristics in all directions. A GFRP composite material exhibits anisotropic properties by providing reinforcement in the tension direction, creating, under certain conditions, more durable structures at lower weights. However, it is necessary to orient the fibers so that the direction of maximum strength of the material used is as close as possible to the direction of maximum stress or equivalent stress introduced by external loading [10]. Otherwise, it is possible that, if the direction of maximum stress is set along a direction which is much different from that of the reinforcement, especially at fatigue stresses, early failure is likely to occur. In order to make the wind turbine blade stronger and stiffer, the construction of the material is made with layers of fibers laid at different angles. Because such a construction consists of different layers, it is often called a sandwich construction [11].

During operation, the wind turbine blade is subjected to the following stresses: bending in two planes, axial stresses and torsion. Axial stresses are produced by the blade’s own weight (to which can be added the weight of the windrows deposited on the blade) and centrifugal forces. These two forces act in the same direction when the blade is in a vertical position below the hub and in opposite directions when the blade is in a vertical position above the hub. The highest stresses are introduced by bending the blade. The stresses produced by axial and bending loading are normal in relation to the cross-section and are often variable in the dynamic regime. Thus, a complex state of stresses arises in the blade, where normal stresses have very high values.

Although attempts are made to transfer the stress to the interior structure, the GFRP composite material is also heavily stressed. Problems arise when dynamic loading changes the direction of the principal stresses, which are no longer directed along the reinforcement fibers [12]. Therefore, the orientation of the reinforcement fibers is of particular importance both for the realization of the strength conditions of the blades and for the economic aspects of material economy. In a study [13], the tensile, compressive and bending characteristics of a GFRP material extracted from turbine blades that have been in operation for 20 years were investigated. It was found that the tensile properties at ordinary and elevated temperatures are almost the same, with better properties being found for samples taken from the length of the wind turbine blade. The highest compressive strength is obtained for samples taken in the transverse direction and the lowest in the blade-length direction and decreases with increasing test temperature. Bending properties are significantly affected in relation to fiber orientation, with the best properties for samples aligned with the blade length and the worst properties for samples with the transverse direction. The use of GFRP materials can be influenced by both temperature variations and operation at low or high temperatures. In the research presented in [14], tensile tests up to 300 °C, compression tests up to 250 °C and shear tests up to 200 °C were performed on GFRP laminates produced by vacuum infusion with a balanced fiber architecture, forming sandwich panel plates. The characteristics obtained by the above tests were strongly affected by the temperature increase. For example, at 200 °C, the shear modulus was 88% lower and compressive strength was 96% lower, both in relation to determinations made at room temperature. To a lesser extent, the tensile strength was affected—40%—and the Young modulus—48%.

Hussain et al. [15] investigated the effects of fiber orientation angle on energy consumption in processing GFRP composite tubes. The effect of fiber orientation on the mechanical behavior of an automotive bumper composite was studied by Nabel Kadum Abd-Ali et al. [16]. The effect of fiber orientation on the mechanical properties and machinability of GFRP composites by milling using shear force analysis was studied by Lakshmankumar Abburi [17]. Kiran Mahadeo Subhedar et al. [18] have been researching carbon-fiber-composite laminates for the effect of their ply configuration, including the number and relative orientation of fiber angles on mechanical properties, and its consistency with the results obtained from simulated data using finite element analysis (FEA) on mechanical properties. Baosheng REN et al. [19,20] describe the effects of fiber orientation angle and fluctuation on the Young’s modulus and tensile strength of so-called full green composites. Cordin, M et al. [21] studied the effect of reinforcement fiber orientation on the mechanical properties of bio-based lyocell-reinforced polypropylene composite. Stanciu et al. [22] present results from the tensile testing of two types of glass-fiber-reinforced polymers (one reinforced with fabric RT500 and the other type reinforced with chopped glass fibers MAT450 and MAT225). Different tensile and elastic characteristics resulted in the two cases. Different characteristics are also observed between tensile and compressive loading.

The experimental study carried out in [23] is aimed at the tensile characterization and failure pattern of glass yarns and GFRP specimens under different loading conditions. The effects of different strain rates as well as temperature on mechanical properties and fracture morphologies are investigated and comparatively discussed.

The composite stress strength criteria also use mechanical characteristics provided by testing the composite material at simple stresses.

Wind turbine blades, which have a complex geometry and are subjected simultaneously to the composed stresses listed above, can be adequately designed by finite element analysis (FEA) [24,25,26,27].

In this paper, a series of determinations of some mechanical characteristics resulting from tensile testing on samples taken from sandwich-structured GFRP were carried out following different directions in relation to the orientation of the fibers. Tensile tests to failure were carried out, from which the characteristics of the stress–strain curve were taken. Tensile tests in the elastic range were carried out on samples taken as above on which strain gauges were bonded in the longitudinal and transverse direction of the samples in order to determine the elastic characteristics. Fractographic SEM analyses were carried out on the resulting fracture sections, differentiating between surfaces and between reinforcement directions.

Experimentally determined mechanical and elastic characteristics were used then for FEA and verification with the Tsai–Hill criterion of a wind turbine blade with a rotor diameter of 4 m, loaded with static forces and moments.

Based on the tests carried out, among the reinforcement solutions adopted, it was determined which is the best solution for use in wind turbine blades. It is taken into account that, during the loading, the direction of the maximum stresses may change, thus unfavorably orienting the fibers. The novelty of this work lies in the fact that both mechanical and elastic characteristics were determined in three different directions with respect to the fiber orientation directions, using the experimental values obtained in the finite element analysis. SEM analyses of the fracture surfaces revealed the character and mode of fracture under tensile stresses.

## 2. Materials and Methods

### 2.1. Materials

Two composite plates reinforced with RT 500 glass fiber and epoxy resin type EPIKOTE MGS LR 385 were made by the lay-up method to determine the material characteristics. The characteristics of the resin are shown in Table 1 [28].

The working method consisted of choosing a non-laminated chipboard of suitable dimensions, which was cleaned, dried and degreased with acetone. In order not to stick to the materials applied to it, three layers of liquid wax-based release agent were applied successively. The drying time for each layer of liquid wax was 20 min and finally, after drying, a polishing process was carried out. In the next step, the resin and hardener were weighed and dosed and then mixed together in plastic pots until homogenized. Each layer was successively impregnated with a textile roller and the excess air between the layers was removed with a metal roller with circular grooves. After the impregnation of all the layers, they were left to cure for 24 h at a temperature of 20°. After complete curing, the board was peeled off the base plate with plastic wedges. As the edges were, as usual, progressively thinned to a width of about 3 cm, the thickness was measured and the outline, which is 4 mm thick, was traced. The thinner part was removed by cutting.

The first composite board is composed of 10 layers, Figure 1a, having the fiber orientation at [0°/90°] from which fifteen specimens were cut in three different directions (5 transverse (TR), 5 longitudinal (LG) and 5 diagonal at 45°), Figure 1b.

The second plate has the same number of layers as the first, but the fiber orientation is at [−45°/0°/+45°/90°], Figure 2a, and from which ten specimens were cut in two directions (5 transverse (TR), 5 longitudinal (LG)), Figure 2b.

The composite plate type [0°/90°] contained 5 pieces of tissue arranged unidirectionally at 0° and another 5 pieces of tissue arranged unidirectionally at 90°. Each piece contained overlapping glass fibers rowing bound together as if there were two layers of unidirectional glass fibers rowing at 0° and 90°. In the end, this results in 10 alternately overlapping unidirectional layers, 5 of which are oriented at 0° the others being oriented at 90°. In the same way, the composite plate [−45°/0°/+45°/90°] was made, the sequence of unidirectional layers being as follows: −45°/+45°/0°/90°/+45°/−45°/90°/0°/−45°/+45°/.

### 2.2. Tensile Tests

To determine the material characteristics, the samples were subjected to a tensile test. In this study, in order to analyze the mechanical behavior of the composites, the samples were tested on a universal testing machine type INSTRON 8801, with load capacity of 100 KN. The test regime was determined by the rate of strain increase at the value of 0.5 mm/min for each specimen until breakage. Due to the low speed of displacement during the test, we have the possibility to observe the deformations while they occurred, obtaining good accuracy in the characteristic curve drawing. The deformations in the direction of stress were measured using a strain gauge with a length between the marks of 25 mm. The tensile tests were carried out according to ASTM D 3518 [29] for specimens with fiber orientation at [45°/0°/+45°/90°], four in the longitudinal and four in the transverse direction. For specimens cut from the plate with fiber orientation at [0°/90°], ASTM D 3039 [30] was followed and four each were obtained in the longitudinal (LG), transverse (TR) and diagonal directions at 45°, Figure 3.

Based on the results obtained after breaking the samples, the stress–strain characteristic curves were plotted. In view of the relatively large scatter of the results obtained and in order to determine some of the characteristic values to be entered into the finite element program, a statistical analysis of the data was also carried out.

To obtain Young’s modulus and Poisson’s ratio for the GFRP composite material of both plates, bi-directional electro-tensometric transducers, type CEA-06-125WT-120, manufactured by Micro-measurements, Raleigh, NC, USA with resistance R = 120 Ω ± 0.35%, were bonded on one specimen each with gauge factor k_G_ = 2.025 ± 0.5%, Figure 4. Under these conditions, the strain gauge in the longitudinal direction will measure the longitudinal specific strain, ε_L_, and the strain gauge on the transverse direction will measure ε_T_.

The mounting of the tensor rosettes was quarter-bridge, and data acquisition was performed using the Vishay P3 two-channel tensor bridge. The experimental results obtained after load application were stored on data files for further processing.

### 2.3. Compression Tests

Compression tests were performed according to ASTM D3410 [31] and a test specimen was cut out of the fiberglass-reinforced composite plate at [0°/90°], Figure 5.

For the compression test, the axial stress direction was parallel to the longitudinal direction of fiber orientation.

### 2.4. Electron Microscopy Analyses (SEM)

In order to highlight the morphology of the experimental samples, two SEM microscopes were used to present the GFRP composite plate directions of the experimental and base material. Additionally, for electrically conductive improvement, a Luxor Au SEM COATER—CT-2201-0144 was used for applying a 7 nm layer of gold on the surface and cross-section of the samples. Two SEM microscopes were used, a SEM FEI Quanta 200 3D microscope (Brno, Czech Republic), using the following parameters: low vacuum mode, LFD (large field detector), HV (high voltage): 20 kV, WD (working distance): 10 mm, similar to previous research [32], and a Vega TESCAN SEM microscope (Brno, Czech Republic) was used, with the following parameters [33]: secondary electrons (SE) detector, electron gun supply: 30 kV, high vacuum and 15.5 mm working distance.

## 3. Results

### 3.1. Mechanical Properties of GFRP under Static Tensile Loading at [0°/90°] and [−45°/0°/+45°/90°]

Figure 6 shows the differences in tensile strength behavior for the cut specimens (LG, TR and at 45°) from the plate with a fiber orientation at [0°/90°] in the overlapped stress–strain curves. Of the four samples taken from the [0°/90°] plate in each of the directions mentioned in Figure 6, only one with higher relevance is shown. Comparing the three graphs, it is observed that the maximum failure stress occurs at the TR flowed specimen, having a maximum value of σ_r_ = 334 MPa. The stress–strain curves for LG and TR show a low strain energy storage, this aspect being specific to materials with elastic viscous behavior. The diagonally cut cross-section stores high strain energy and this aspect highlights an elasto-plastic behavior. This behavior is due to the fact that the stress occurs in a direction along which there are no reinforcing fibers. Analyzing the three characteristic stress–strain curves in the TR, LG and diagonal directions, it can be seen that between the elastic limit and the material’s breaking zone there is a first portion that indicates the interlaminar behavior and response of the laminate during stress. Point 1 marked in all three cases also represents the first interlaminar microcrack. As the force increased, more fibers failed. The loads were transferred to other uninterrupted fibers, which in turn took up more energy; this is visible at point 2 on all three curves.

The three directions (TR, LG and diagonal at 45°) of the plate [0°/90°] and two directions (TR and LG) of the plate [−45°/0°/+45°/90°] were analyzed for failure type and failure mode. These aspects complied with the stress-test failure codes/typical modes of ASTM D3039. In the case of the specimens from the [0°/90°] plate on the LG and TR direction, they comply with the DGM (edge delamination gage middle) tensile test failure codes, but in the 45 direction we find the AGM (2)—angled gage middle code, Figure 7.

In what follows, a statistical analysis is presented for all sample sets from the plate with fiber orientation at [0°/90°], Table 2, Table 3 and Table 4. The formulas used were the same for all three cases.

The arithmetic mean of these four data points is the average ultimate tensile stress:(1)$${\bar{\sigma }}_{\mathrm{UTS}}=\frac{1172.8}{4}=293.2\;\mathrm{MPa}$$

The sample standard deviation is:(2)$$S=\sqrt{\frac{\sum {\left(\sigma_{\mathrm{UTS}}-\sigma_{\mathrm{UTS}}\right)}^{2}}{n-1}}$$
$$S=\sqrt{\frac{80.01}{3}}=5.16\;\mathrm{MPa}$$

The coefficient of variation CV is defined as the ratio of the standard deviation S to the mean
(3)$$\mathrm{CV}=\frac{S}{\sigma_{r}}$$
(4)$$\mathrm{CV}=\frac{5.16}{293.2}100=1.75\%$$

In Figure 8, two stress–strain curves are overlapped, for two more relevant specimens of those cut on the same direction. The specimens were cut from the [−45°/0°/+45°/90°] plate. Analyzing the graphs, it can be seen that the maximum failure stress occurs in the LG cut specimen with a maximum value of σ_r_ = 200 MPa. The appearance of the two curves in the area marked 1 shows that the first interlaminar microcrack occurs at a stress of about 50 MPa.

For the plate with orientation at [−45°/0°/+45°/90°] on the TR direction, the code XGM- explosive gage middle was identified and on the LG direction the failure mode was AGM (1)- angled gage middle, Figure 9.

In this case too, a statistical analysis of the tensile test results was performed on all specimens, Table 5 and Table 6.

Figure 10 plots the normal stress (σ)/longitudinal specific strain (ε_L_) coordinate plots of the longitudinal modulus of elasticity, E (Young’s modulus), for the specimens in the plate with orientation at [0°/90°] and [−45°/0°/+45°/90°]. The Young’s modulus was determined as the slope of the line of approximation of the graph plotted on the coordinates of normal stress (σ)/longitudinal specific strain (ε_L_), through the points determined from the records of the force values by the Instron 8801 machine calculator and the longitudinal specific strains. Points located at low force values (influenced by the initial clamping processes in the machine bins) and points located at high force values, over which no strain recordings were made with the P3 bridge, were removed from the graph, because detachments of the tensiometer rosette and distortions of the strain recordings occurred.

In Figure 11, plots in the coordinates of transverse specific strain (ε_T_)/longitudinal specific strain (ε_L_) for determining the Poisson’s ratio ν, on the specimens in the plate with orientation at [0°/90°] and [−45°/0°/+45°/90°] are shown. The slope of the approximation line represents the coefficient of transverse Poisson’s shrinkage.

The results of the static tensile tests confirmed that the cut specimens (TR, LG) from the [0°/90°] plate definitely showed much higher strength. The elastic properties in terms of longitudinal modulus of elasticity values are about twice as high on [0°/90°].

In [34], shear tests were presented with the Iosipescu method ASTM D5379, on specimens taken from the same composite material and τ_12_ = 54 MPa was obtained, on a specimen reinforced at [0°/90°]. A comparison between the shear-test methods for composite materials shows that the Iosipescu method is considered the most accurate [35].

### 3.2. Compression Test

Figure 12 plots the compressive stress–strain curve for a specimen cut from the [0°/90°] plate. The failure section was approximately at 45° to the direction of the force and a σ_c_ = 342 MPa was obtained.

### 3.3. SEM Surface at [0°/90°]

In the electron microscopy analyses, the determinations were performed on the representative samples with the highest stresses obtained from the mechanical tests. The aim was to highlight clear aspects of fiber breakage and characteristic areas. Figure 13 and Figure 14 present the SEM images of the GFRP base material’s surface and cross-section, as well as the tensile strength analysis specimens for the three directions (LG, TR and 45°). Figure 13a shows the surface of the base material from the morphological point of view. The surface exhibits a compact and no microcracks are observed. Figure 13b shows the 10 polymer layers of the GFRP cross-section, used in wind turbine blade construction.

Figure 14 displays the cross-sectional images from which the samples were collected. Thus, Figure 14a presents a fiber break in the transversal direction, whereas Figure 14b highlights a fiber break at 45°; these two fiber breaks have a major damage aspect. Figure 14c shows a delamination at the intermediate layers, with the fracture exhibiting moderate behavior.

Figure 15, Figure 16 and Figure 17 exhibits the surface morphology of the examined samples in all three stress directions. In all three examples, a comparable brittle-type shattering process is seen in the fiber-resin matrix. The different powers of magnification demonstrate the fracture behavior and appearance of the polymer pull-out fibers at the interface of the resin. Figure 15 highlights a more compact and equally dispersed model of these fibers, whereas Figure 16 and Figure 17 demonstrate a more significant dislocation.

### 3.4. SEM Surface at [−45°/0°/+45°/90°]

Figure 18 presents the SEM images of the GFRP base material’s surface and cross-section; Figure 18a shows the surface of the base material from the morphological point of view. The surface exhibits a compact character and some small microcracks are observed. The Figure 18b shows the structure of the composite material consisting of the base matrix and epoxy resin—EPIKOTE.

The following are electron microscopy analyses, in two directions (LG and TR), on tensile specimens with pronounced layer damage. Figure 19 displays the cross-sectional images of the tested samples (TR and LG). Figure 19a presents a fiber break in the transversal direction, whereas Figure 19c highlights a fiber break at longitudinal orientation, these two fiber breaks have a major damage aspect. Figure 19d shows a partial delamination at the intermediate layers, with the fracture exhibiting moderate behavior.

The surface morphology of the studied samples in two stress directions is shown in Figure 20 and Figure 21. In each of the two cases, the fiber-resin matrix exhibits a similar brittle-type fracture mechanism. Figure 20 illustrates a more compact and uniformly morphology model of these fibers, while Figure 21a,b exhibits a more pronounced dislocation.

### 3.5. Finite Element Analysis of a Wind Turbine Blade

A structural simulation was made for a wind turbine blade, which is one of the three blades equipping a 4 m diameter rotor. The power of the turbine is about 3 kW. The blade consists of three elements (Figure 22): the current blade (the outer shell, made of GFRP), a stiffening beam and a hub inside, both made of steel. The wind turbine blade has a NACA aerodynamic profile. For FEA, the elastic characteristics of GFRP as experimentally determined were used.

The finite element analysis (FEA) was performed in the ANSYS Academic R17.2 software. The three components of the blade were discretized individually with three-dimensional elements, of a tetrahedral shape. For all components, a curvature size function, with a curvature normal angle of 24°, was used, in order to follow the areas with a more pronounced curvature, creating a finer discretization network in their vicinity. The mesh has a number of 1,309,769 nodes and 768,476 elements (Figure 23). The three components of the blade were solidarized among themselves. The experimentally determined elastic characteristics of GFRP were used for FEA. Horizontal axis wind turbines (HAWTs) with three blades are designed for a tip-speed ratio between six and nine:(5)$$\lambda =\frac{\omega R}{U}$$
where *ω* is angular velocity, *R* is radius (between the center of the rotor and the tip of the blade) and *U* is the incident wind velocity. With the increase of tip-speed ratio, the stresses in the blade due to aerodynamic forces and centrifugal force increase proportionally to the angular velocity and, respectively, to the square of the angular velocity, due to the reduction of the aerodynamic chord [36].

In the MEXICO project, a series of experiments and numerical simulations were carried out for a wind turbine rotor with three blades with a diameter of 4.5 m, at different wind velocity and rotational speeds, with a tip-speed ratio of 6.7. Following the experiments, a series of aerodynamic parameters were determined, including the aerodynamic forces (normal force or thrust force and tangential force) and the torque [37]. In this structural simulation, *U* = 15 m/s and = 6.7 was adopted for the blade of a rotor with a diameter of 4 m, which has three blades. With these values, Equation (4) results in an angular velocity of 50.25 rad/s, which corresponds to a rotational speed of 480 rpm. The mass of the three components of the blade and the position of the center of mass of the blade was determined in CATIA, considering the steel density of 7860 kg/m^3^ and the GFRP density of 2000 kg/m^3^. This results in a total mass of the blade of 11 kg and, respectively, a weight of 108 N. With these values, a centrifugal force of 12,972 N was calculated. These loads are distributed on the surface of the blade, but in the FEA they were considered applied to the tip of the blade. This last case of loading is more unfavorable (it generates higher stresses in the blade). Consequently, the blade calculated with the loads applied to its tip will better resist (with a higher safety coefficient) in the case of distributed loads.

Due to almost the same overall dimensions of the wind turbine rotor, the other loads were considered approximately as in MEXICO project, for a wind velocity of 14.9 m/s, a rotational speed of 424 rpm and a tip-speed ratio of 6.7; the normal force is 1500 N, tangential force 80 N and the torque 300 Nm. Thus, it is considered a fixed support for the outer surface in the blade-hub area (A), with the application of the following loads on the tip of the blade (Figure 24):-Torque (B) around the Z axis, with a value of 300 Nm;-Tensile force (C) oriented in the positive direction of the Z axis, which represents the sum of the centrifugal force and the gravitational force, equal to 13,080 N (this value corresponds to the position of the blade below the rotor hub, in the vertical direction);-A force oriented in the positive direction of the Y axis, which represents the normal component of the aerodynamic force (D), with a value of 1500 N;-The tangential component of the aerodynamic force (E), oriented in the positive direction of the X axis, with a value of 80 N.

After the FEA, the stress distribution on the blade was obtained. The distributions of the von Mises stresses on the extrados and, respectively, on the intrados of the blade are shown in Figure 25. It is observed that the finite elements (FE) on the outer surface of the blade are more stressed on the intrados compared to the extrados of the blade. The maximum von Mises stress of 110.48 MPa is located on the extrados of the blade, near its inner surface, at the coordinate point (X: 11.13 mm, Y: 18.3 mm, Z = 1552.1 mm), relative to the coordinate system located at the blade bottom. This can be seen in Figure 26, where the cross-section of the blade and view from the hub is presented. The principal stresses in FEA with the highest equivalent (von Mises) stress are presented in Table 7. It is observed that the maximum stress in absolute value is produced by compression.

Another highly stressed point is located on the opposite side of the one above (Figure 27). The principal stresses at this point are presented in Table 8. This point is located on the intrados of the blade and has the coordinates (X: 10.52 mm, Y: −23.2 mm, Z = 1552.1 mm).

### 3.6. Turbine Blade Calculation at Static Loading

The von Mises criterion is used especially for the calculation of homogeneous and isotropic materials in complex states of stress. Other failure criteria are used for the calculation of composite materials: Tsai–Hill, Tsai–Wu, Hoffman, Christensen, etc. [38,39,40,41]. The Tsai–Hill criterion will be used to calculate the WRB. The experimentally determined tensile and compression strength of GFRP were used for turbine blade calculation. For three principal stresses, this criterion is reduced to:(6)$$\frac{1}{2\sigma_{T}\sigma_{C}}\left[{\left(\sigma_{1}-\sigma_{2}\right)}^{2}+{\left(\sigma_{2}-\sigma_{3}\right)}^{2}+{\left(\sigma_{3}-\sigma_{1}\right)}^{2}\right]=1$$

In [42], a new elastoplastic continuum damage model, based on Puck’s theory, is presented. The model is intended for the plane state of stress. It is capable of describing non-linearity due to the irreversible strain in the matrix but cannot be used for composites with textile plies. Taking into account the particularities of the tested material, the Tsai–Hill criterion will be used to calculate the WTB analyzed above. For a triaxial state of stress, the composite material breaks when Equation (5) is fulfilled, and it resists when the left-hand side of Equation (5) is less than 1. It can be seen that when σ_T_ = σ_C_, the Tsai–Hill criterion degenerates into von Mises. Next, the Tsai–Hill criterion is applied with the main stresses from Table 7 and Table 8, respectively, using the tensile and compressive stresses at break, which will be taken into account:$$\frac{1}{2\cdot 293.2\cdot 342}\left[{\left(1.0-5.2\right)}^{2}+{\left(-5.2+111.2\right)}^{2}+{\left(-111.2-1.0\right)}^{2}\right]=0.23<1$$
$$\frac{1}{2\cdot 293.2\cdot 342}\left[{\left(88.85-16.7\right)}^{2}+{\left(16.7+1.28\right)}^{2}+{\left(-1.28-88.85\right)}^{2}\right]=0.12<1$$

For the most dangerous state of the stress coefficient of safety for static loads is c = 1/0.23 = 4.35. It results from this that the GFRP material of the blade resists at the proposed static loads, considering both stress states presented in Table 7 and Table 8, which turned out to be the most dangerous, according to AEF. Although the blade can still resist at higher static loads, it must be considered that it must also resist also at higher wind speeds (up to the dangerous speed, that of locking the rotor) at fatigue, dynamic loads, creep and buckling, as well as in harsh environmental conditions.

## 4. Conclusions

In this study, a type of material used in the manufacture of wind turbine blades was presented and tested. The material used was a glass-fiber-reinforced composite GFRP, and following static tensile, compression and electron microscopy tests, the following conclusions were drawn:In tensile tests, the material’s behavior is anisotropic due to the different tensile stresses occurring in all directions on the two-plate GFRP;Samples that were cut from the [0°/90°] plate in the TR direction had the highest resistance to the force applied parallel to the direction of force application. This is mainly caused by the alignment of the reinforcing fibers;Tensile loading of the sample cut at 45° from the [0°/90°] plate resulted in much lower ultimate tensile strength values than those obtained by loading in the other two directions. On the other hand, the elongation, and therefore strain, on this sample was much higher than on the other two;A different value for Young’s modulus was obtained for the sample loaded and cut at 45° relative to the other two directions, from the plate with the fibers oriented [0°/90°];For the [0°/90°] plate, a much different value of the Poisson ratio was obtained in the 45° loading direction, compared to the other two directions. This difference, as for the difference in Young’s modulus, is explained by the fact that there are no fibers in the loading direction that directly oppose the stress. For the [0°/90°] plate, a much different value of the Poisson ratio was obtained in the 45° loading direction, compared to the other two directions. This difference, as the difference in the Young’s modulus, is explained by the fact that there are no fibers in the loading direction that directly oppose the stress;The values for the Poisson ratio for the [−45°/0°/+45°/90°] plate obtained on the two loading directions were similar. However, they are also found to be similar to the value obtained for the [0°/90°] plate cut and loaded at 45°. This is explained by the fact that the two directions in which the plate was cut did not contain fibers that were placed exactly in the direction of the load;For the [0°/90°] plate orientation, a fiber break in the transversal direction has been highlighted. Additionally, a delamination at the intermediate layers, with the fracture exhibiting moderate behavior, has been observed. For the [−45°/0°/+45°/90°] transversal plate, a more compact and uniform morphology model of these fibers has been shown, while for the longitudinal plate a more pronounced dislocation has been exhibited;Using ANSYS Academic R17.2 software, a FEA was undertaken for a wind turbine blade with a diameter of 4 m, considering the static loads applied to the tip of the blade. For this purpose, the elastic characteristics of the material were considered;Using the Tsai–Hill criterion and the mechanical characteristics of the material, the turbine blade was checked for the most dangerous stress states provided by the FEA. After this verification, it turns out that the blade can take on static loads higher than those considered in the article. However, it must be taken into account that the blade must occasionally withstand higher rotational speed, as well as fatigue, creep and warping, and harsh environmental conditions.

Consequently, the study carried out in this paper shows that the arrangement of the fibers in relation to the direction of the maximum stress is very important. The stress directions in operation are relatively random and the direction of the maximum stress can differ substantially, for example, in relation to the wind direction. If we consider the material [0°/90°] and if the direction of maximum stress were at 45°, there would be a rapid deterioration of the wind turbine blades due to both a decrease in tensile characteristics and a substantial change in Poisson’s ratio with respect to the other two directions, [0°/90°]. Particular attention should be paid to the orientation of the fibers, especially in the areas of maximum stress revealed by the finite element analysis. In this work, tests were also carried out on the [−45°/0°/+45°/90°] plate, from which significant improvements were found in terms of the much smaller variation of the characteristics in relation to the direction of fiber orientation. Here again it should be taken into account that 10 layers can still be used, leaving fewer layers with an orientation at 90°. Considering that the weight of the wind turbine blade should be as low as possible, adding more fibers is not an option. As a result, at a minimum volume of fibers used, maximum strength should be achieved, and this is achieved through proper fiber orientation.

Future research should focus on:-Optimizing the volume of fibers added in relation to their orientation;-Determinations similar to those carried out in this paper on other types of fiber arrangements;-Measurements of the turbine blade at full or reduced scale to determine the variation of the direction of the maximum normal and shear stresses in relation to different ratios of tensile and torsional loading.

## Figures and Tables

**Figure 1 polymers-15-00861-f001:**
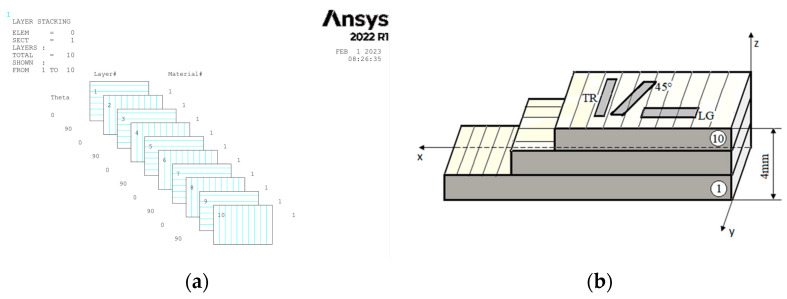
Configuration of layers of composite [0°/90°] (**a**) and directions for sampling from the GFRP plate (**b**).

**Figure 2 polymers-15-00861-f002:**
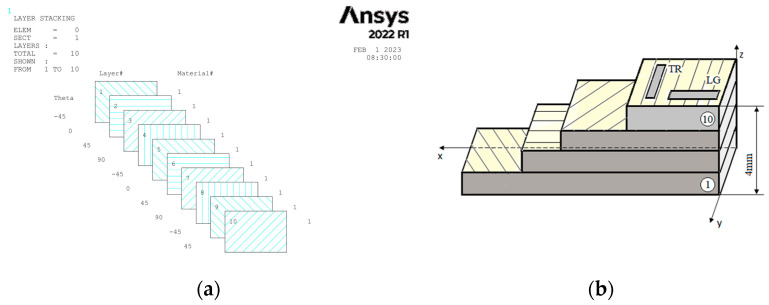
Configuration of layers of composite [−45°/0°/+45°/90°] (**a**) and directions for sampling from the GFRP plate (**b**).

**Figure 3 polymers-15-00861-f003:**
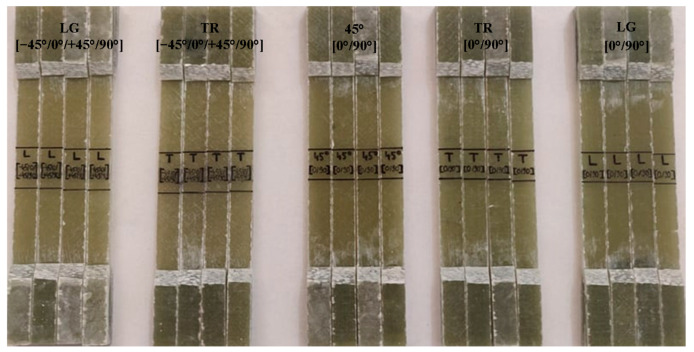
All specimens cut and subjected to tensile stress.

**Figure 4 polymers-15-00861-f004:**
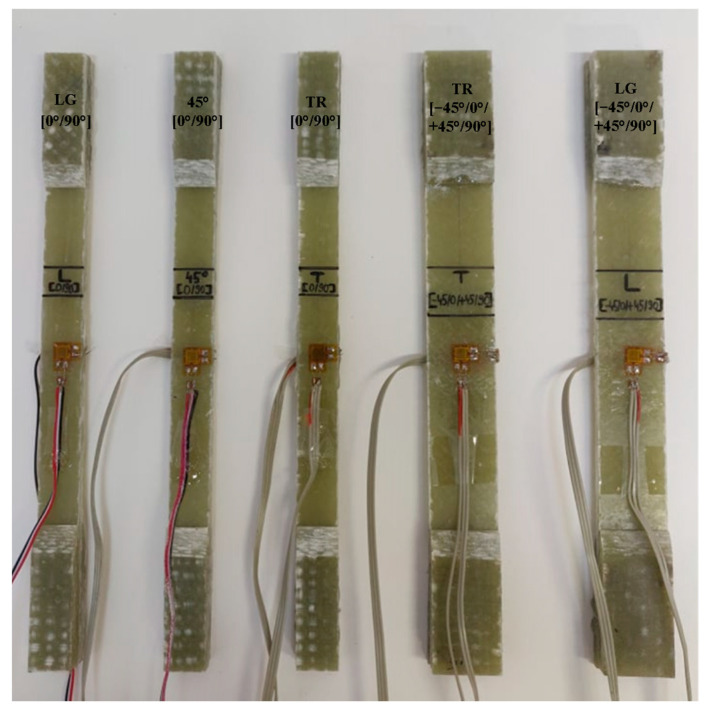
Installation of bidirectional rosettes.

**Figure 5 polymers-15-00861-f005:**
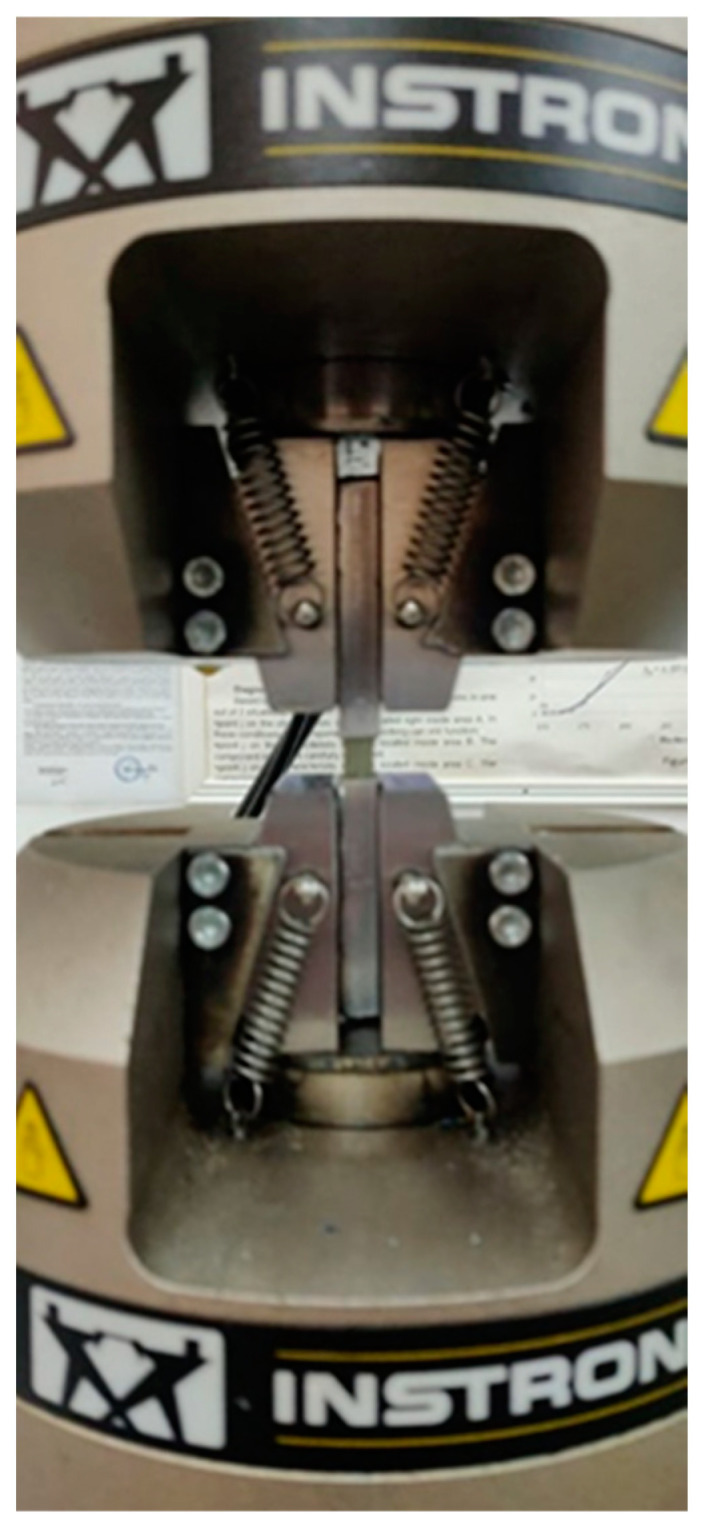
Specimens of compression tests.

**Figure 6 polymers-15-00861-f006:**
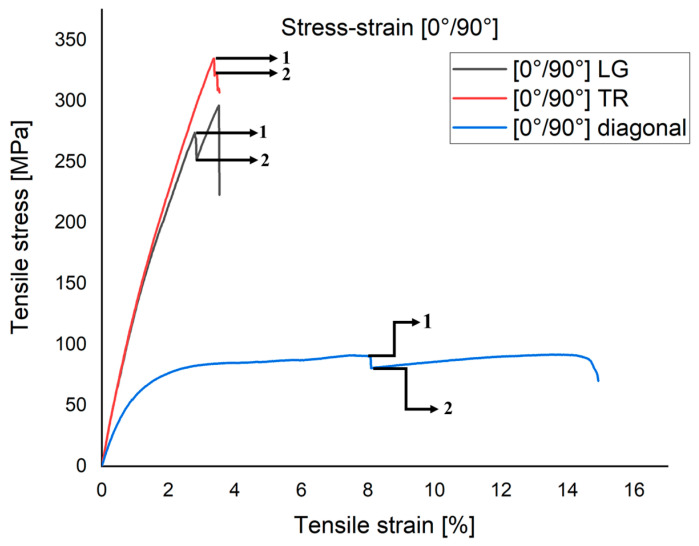
Characteristic curve stress–strain for three sample at [0°/90°]: (1) the first crack initiation point; (2) the second crack initiation point. Point 1 marked in all three cases also represents the first interlaminar microcrack. The loads were transferred to other uninterrupted fi-bers, which in turn took up more energy; this is visible at point 2 on all three curves.

**Figure 7 polymers-15-00861-f007:**
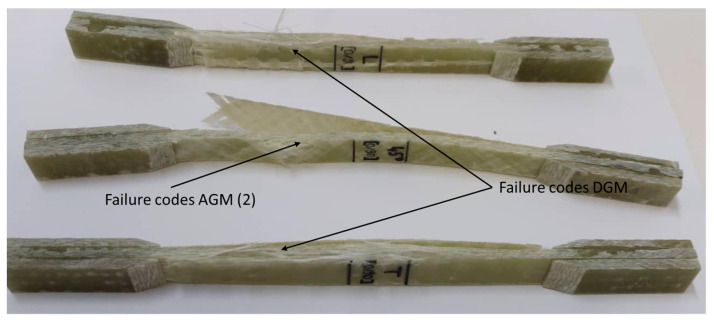
Failure type and failure mode at specimen TR, LG and diagonal at 45° of the plate [0°/90°].

**Figure 8 polymers-15-00861-f008:**
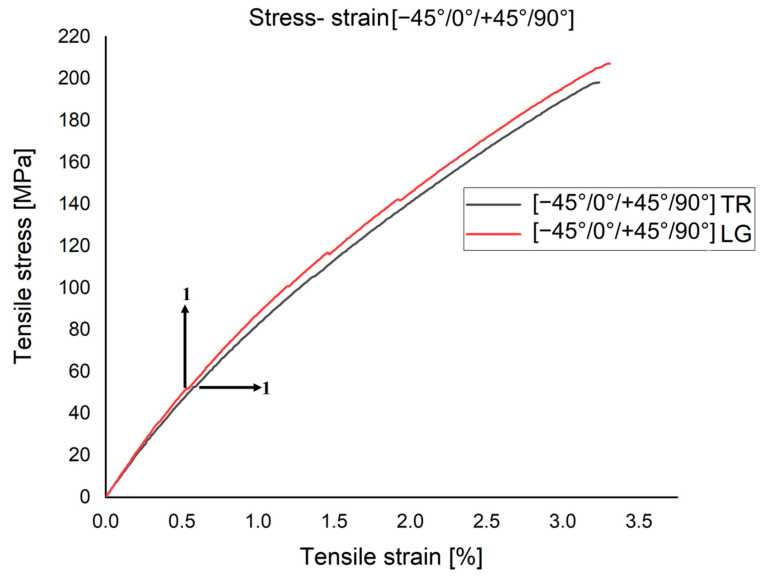
Characteristic stress–strain curve for three samples at [−45°/0°/+45°/90°]: (1) the first interlaminar microcrack. Point 1 marked the first interlaminar microcrack.

**Figure 9 polymers-15-00861-f009:**
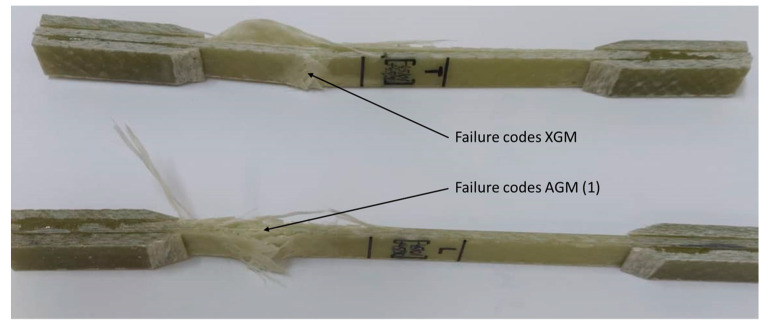
Failure type and failure mode at specimen TR, LG of the plate at [−45°/0°/+45°/90°].

**Figure 10 polymers-15-00861-f010:**
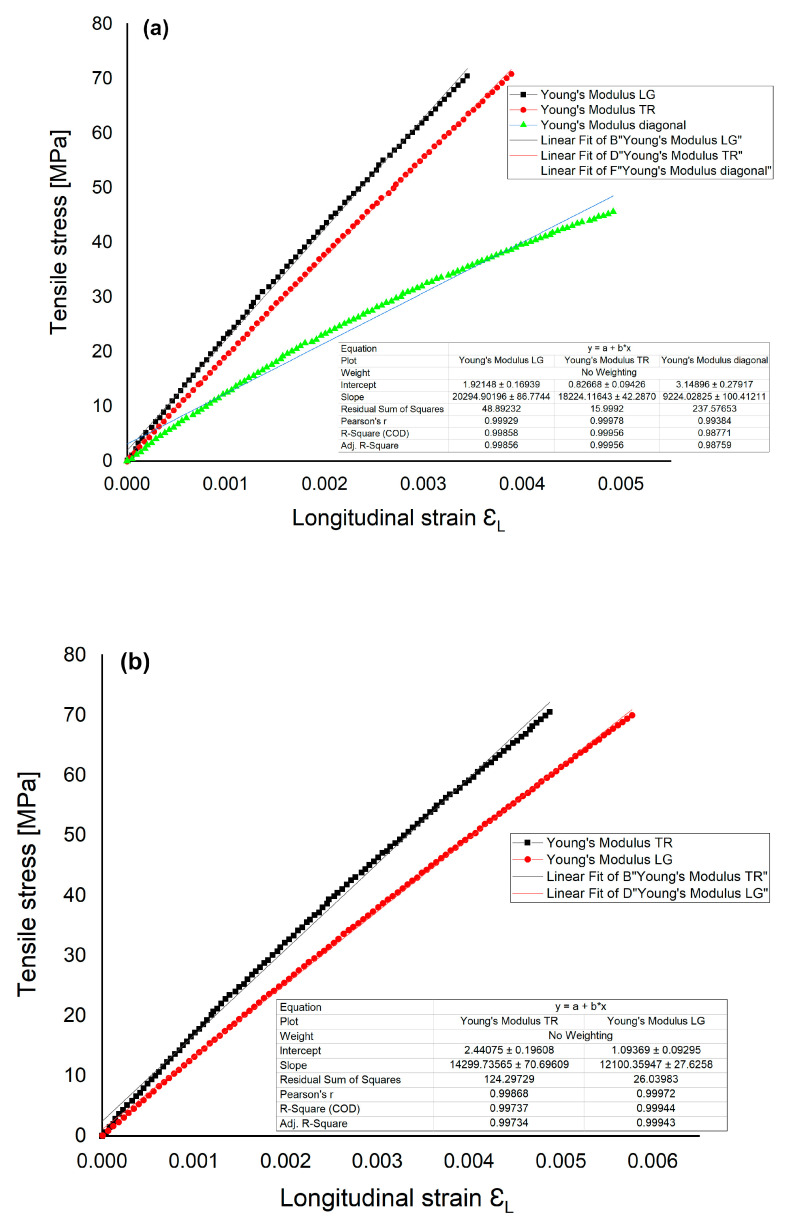
Stress variation with longitudinal strain-approximation lines for longitudinal elasticity modulus at [0°/90°] (**a**) and [−45°/0°/+45°/90°] (**b**).

**Figure 11 polymers-15-00861-f011:**
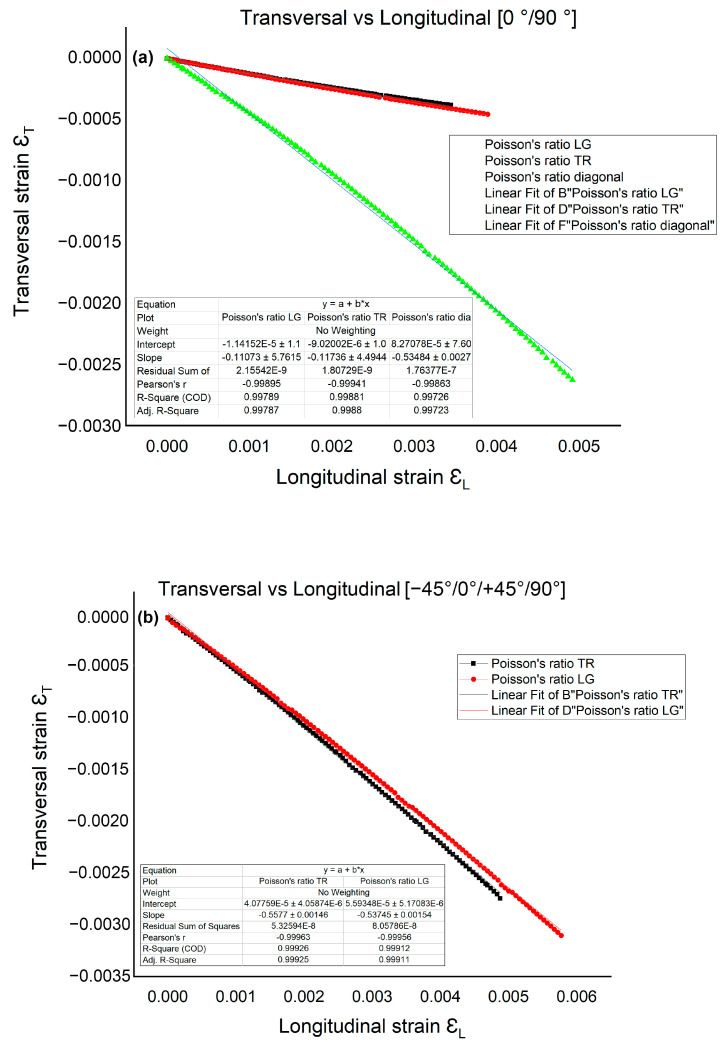
Longitudinal strain-transversal strain curve at [0°/90°] (**a**) and [−45°/0°/+45°/90°] (**b**).

**Figure 12 polymers-15-00861-f012:**
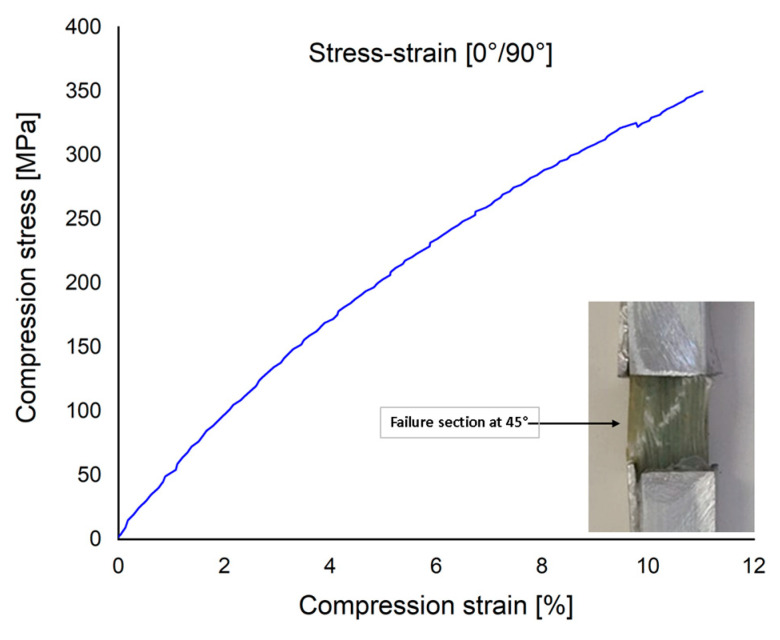
The stress–strain curve of compression tests at [0°/90°].

**Figure 13 polymers-15-00861-f013:**
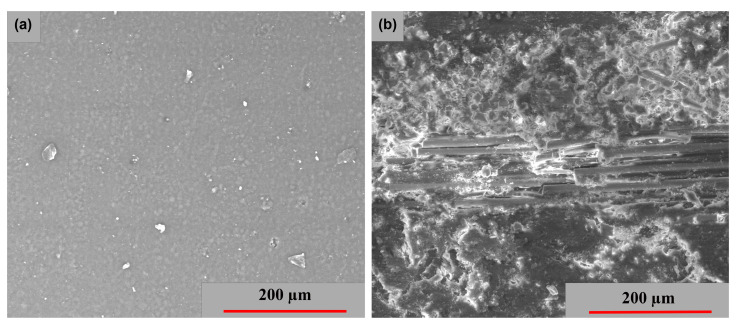
SEM images of the base material—GFRP: (**a**) surface image and (**b**) cross-section image.

**Figure 14 polymers-15-00861-f014:**
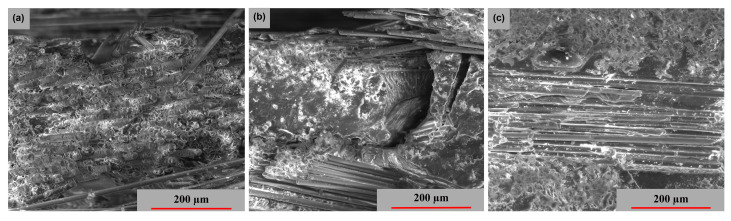
Cross-section SEM images of GFRP specimens in the 3 stress directions: (**a**) transversal, (**b**) oriented at 45° and (**c**) longitudinal.

**Figure 15 polymers-15-00861-f015:**
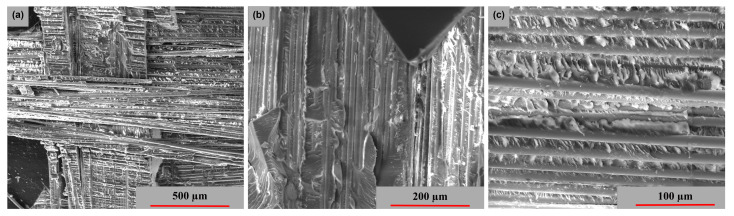
Surface SEM images of transversal [0°/90°] GFRP sample: (**a**) 200×, (**b**) 500× and (**c**) 1000×.

**Figure 16 polymers-15-00861-f016:**
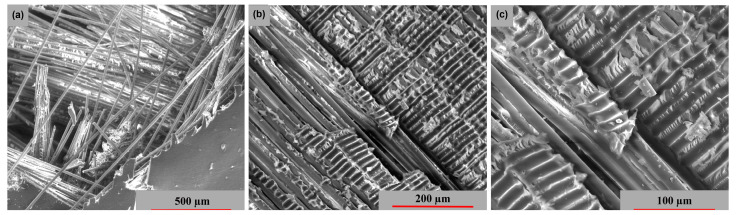
Surface SEM images of oriented at 45° GFRP sample: (**a**) 200×, (**b**) 500× and (**c**) 1000×.

**Figure 17 polymers-15-00861-f017:**
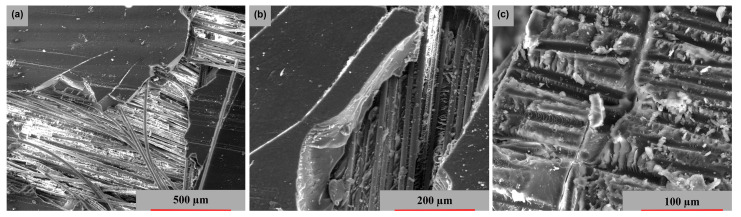
Surface SEM images of longitudinal [0°/90°] GFRP sample: (**a**) 200×, (**b**) 500× and (**c**) 1000×.

**Figure 18 polymers-15-00861-f018:**
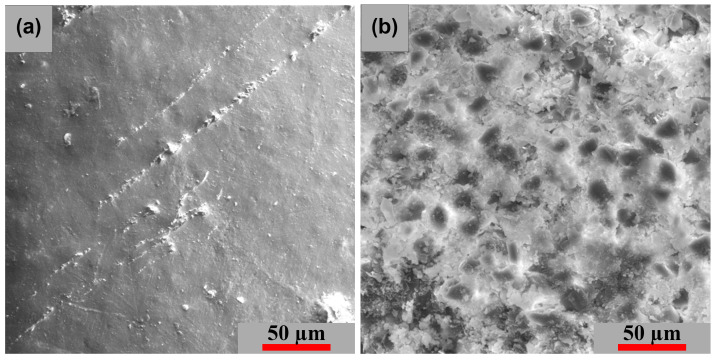
SEM images of the base material—GFRP: (**a**) surface image and (**b**) cross-section image.

**Figure 19 polymers-15-00861-f019:**
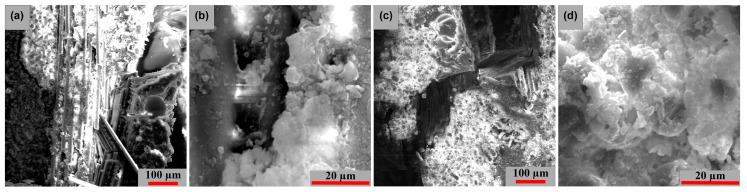
Cross-section SEM images of GFRP specimens in the 2 stress directions: (**a**) 200× (TR), (**b**) 2000× (TR), (**c**) 200× (LG) and (**d**) 2000× (LG).

**Figure 20 polymers-15-00861-f020:**
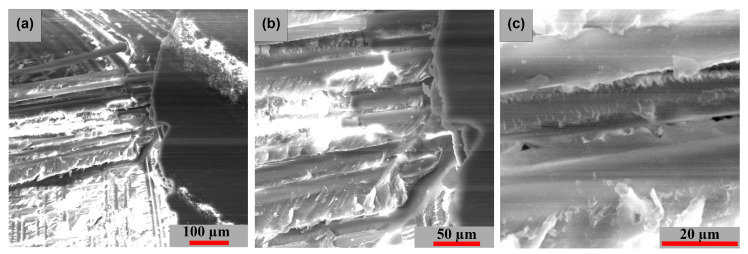
Surface SEM images of transversal [−45°/0°/+45°/90°] GFRP sample: (**a**) 200×, (**b**) 500× and (**c**) 2000×.

**Figure 21 polymers-15-00861-f021:**
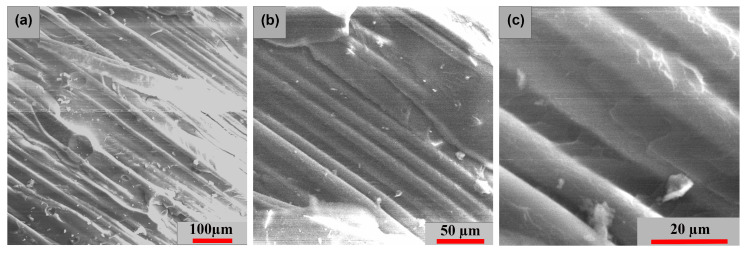
Surface SEM images of longitudinal [−45°/0°/+45°/90°] GFRP sample: (**a**) 200×, (**b**) 500× and (**c**) 2000×.

**Figure 22 polymers-15-00861-f022:**
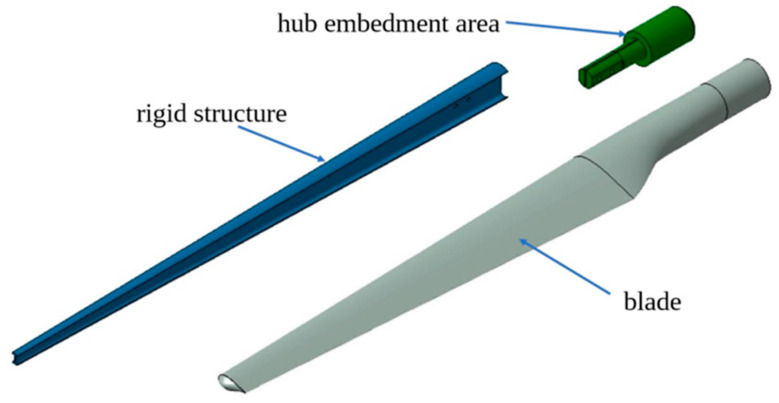
The component elements of the blade.

**Figure 23 polymers-15-00861-f023:**
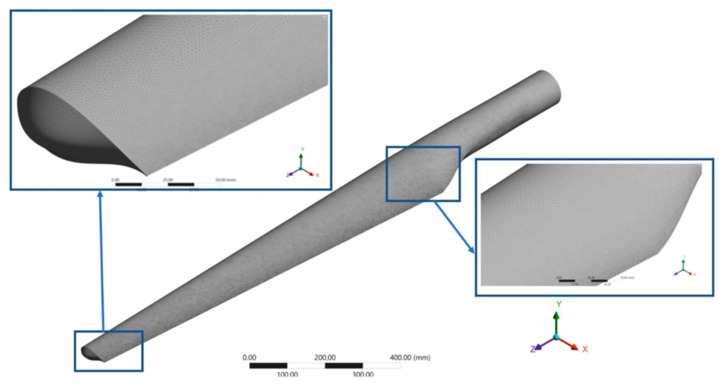
The discretization mesh of the blade.

**Figure 24 polymers-15-00861-f024:**
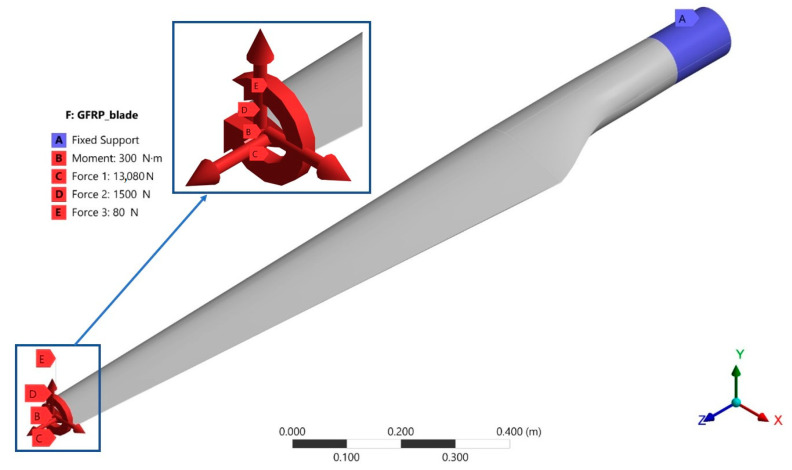
Load status on the blade embedded in the hub area.

**Figure 25 polymers-15-00861-f025:**
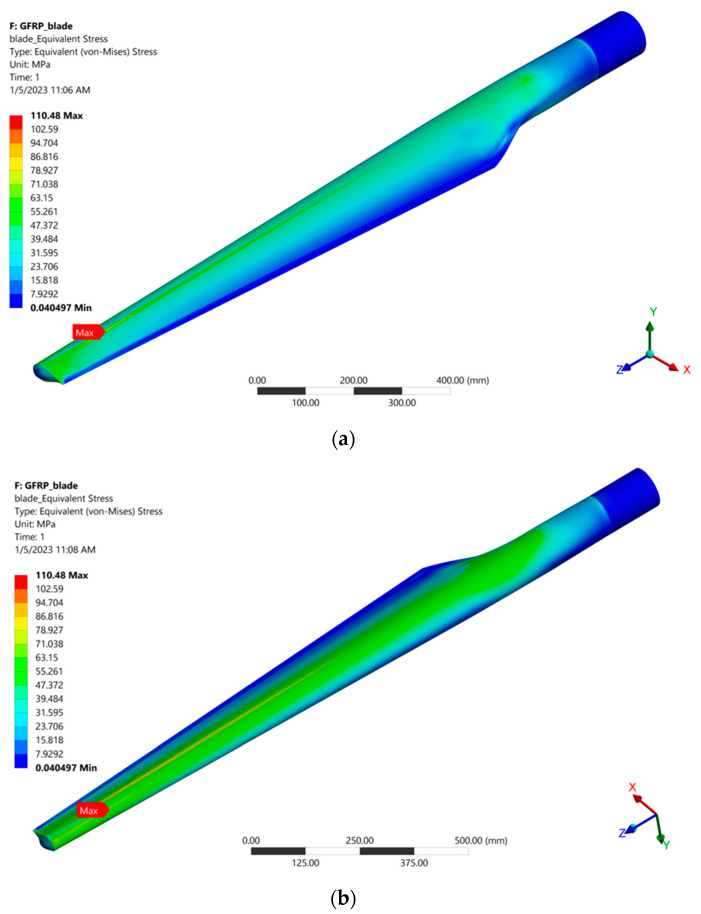
The distributions of the von Mises stress: on the extrados (**a**) and, respectively, on the intrados (**b**) of the blade.

**Figure 26 polymers-15-00861-f026:**
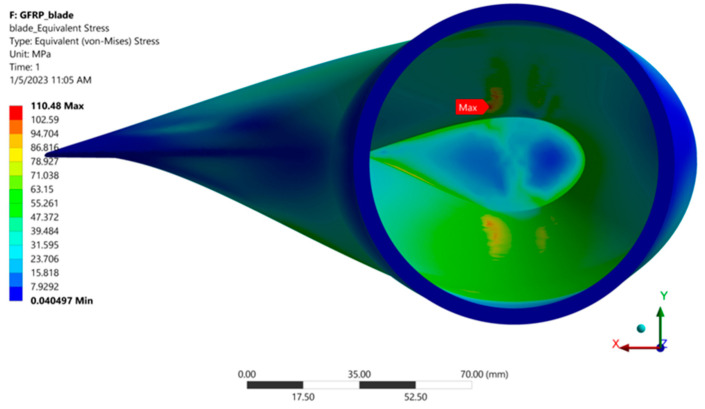
The maximum von Mises stress in the blade (view from the hub).

**Figure 27 polymers-15-00861-f027:**
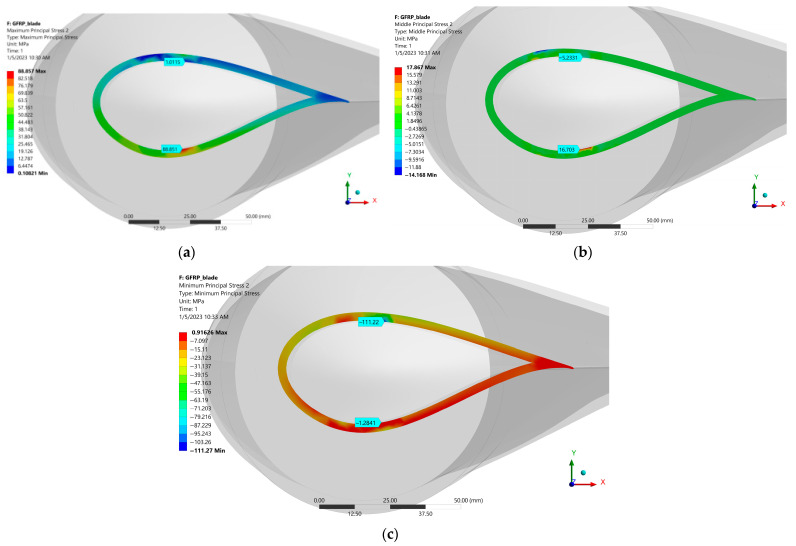
Principal stresses at the two points presented above: (**a**) Maximum Principal Stress, (**b**) Middle Principal Stress and (**c**) Minimum Principal Stress.

**Table 1 polymers-15-00861-t001:** Characteristics of epoxy resin type EPIKOTE MGS LR 385.

Properties	Units	Value
Density	[g/cm^3^]	1.20
Viscosity	[mPa·s]	700–1050
Flexural strength	[N/mm^2^]	120–130
Modulus of elasticity	[kN/mm^2^]	3.3–3.6
Tensile strength	[N/mm^2^]	75–85
Compressive strength	[N/mm^2^]	120–140
Elongation of break	[%]	6–8
Impact strength	[KJ/m^2^]	45–60
Water absorption at 23 °C in 24 h	[%]	0.01

**Table 2 polymers-15-00861-t002:** Standard deviation for ultimate tensile stress σ_UTS_, (longitudinal oriented at [0°/90°]).

Sample No.	σ_UTS_ [MPa]	${\bar{\sigma }}_{UTS}$ [MPa]	Deviations from the Mean $\sigma_{r}-{\bar{\sigma }}_{r}$ [MPa]	${\left(\sigma_{r}-{\bar{\sigma }}_{r}\right)}^{2}$ [MPa]	Standard Deviation S [MPa]	Coefficient of Variation CV [%]
1	296.03	293.2	2.83	8.00	5.16	1.75
2	294.93		1.73	2.99		
3	296.37		3.17	10.04		
4	285.52		−7.68	58.98		
Σ=	1172.8			80.01		

**Table 3 polymers-15-00861-t003:** Standard deviation for ultimate tensile stress σ_UTS_, (transverse oriented at [0°/90°]).

Sample No.	σ_UTS_ [MPa]	${\bar{\sigma }}_{UTS}$ [MPa]	Deviations from the Mean $\sigma_{r}-{\bar{\sigma }}_{r}$ [MPa]	${\left(\sigma_{r}-{\bar{\sigma }}_{r}\right)}^{2}$ [MPa]	Standard Deviation S [MPa]	Coefficient of Variation CV [%]
1	327.47	324.19	3.28	10.75	8.92	2.75
2	314.27		−9.92	98.04		
3	320.17		−4.02	16.16		
4	334.87		10.68	114.06		
Σ=	1296.7			239.01		

**Table 4 polymers-15-00861-t004:** Standard deviation for ultimate tensile stress σ_UTS_, (diagonal oriented 45° at [0°/90°]).

Sample No.	σ_UTS_ [MPa]	${\bar{\sigma }}_{UTS}$ [MPa]	Deviations from the Mean $\sigma_{r}-{\bar{\sigma }}_{r}$ [MPa]	${\left(\sigma_{r}-{\bar{\sigma }}_{r}\right)}^{2}$ [MPa]	Standard Deviation S [MPa]	Coefficient of Variation CV [%]
1	91.82	89.70	2.12	4.49	2.1	2.35
2	89.72		0.02	0.004		
3	90.46		0.76	0.57		
4	86.82		−2.88	8.29		
Σ=	358.82			13.35		

**Table 5 polymers-15-00861-t005:** Standard deviation for ultimate tensile stress σ_UTS_, (longitudinally oriented at [−45°/0°/+45°/90°]).

Sample No.	σ_UTS_ [MPa]	${\bar{\sigma }}_{UTS}$ [MPa]	Deviations from the Mean $\sigma_{r}-{\bar{\sigma }}_{r}$ [MPa]	${\left(\sigma_{r}-{\bar{\sigma }}_{r}\right)}^{2}$ [MPa]	Standard Deviation S [MPa]	Coefficient of Variation CV [%]
1	196.17	200.1	−3.93	15.44	5.95	2.97
2	202.62		2.52	6.35		
3	207.28		7.18	51.55		
4	194.35		−5.75	33.06		
Σ=	800.42			106.4		

**Table 6 polymers-15-00861-t006:** Standard deviation for ultimate tensile stress σ_UTS_, (transversely oriented at [−45°/0°/+45°/90°]).

Sample No.	σ_UTS_ [MPa]	${\bar{\sigma }}_{UTS}$ [MPa]	Deviations from the Mean $\sigma_{r}-{\bar{\sigma }}_{r}$ [MPa]	${\left(\sigma_{r}-{\bar{\sigma }}_{r}\right)}^{2}$ [MPa]	Standard Deviation S [MPa]	Coefficient of Variation CV [%]
1	184.17	188.44	−4.27	18.23	9.12	4.83
2	196.87		8.43	71.06		
3	180.45		−7.99	63.84		
4	198.28		9.84	96.82		
Σ=	753.77			249.95		

**Table 7 polymers-15-00861-t007:** The principal stresses in the FEA with the highest equivalent (von Mises) stress.

Equivalent (von Mises) Stress	110.48 MPa
Maximum Principal Stress	1.01 MPa
Middle Principal Stress	−5.23 MPa
Minimum Principal Stress	−111.22 MPa

**Table 8 polymers-15-00861-t008:** The main stresses in the area where the maximum von Misses stress was obtained (on the extrados of the blade) and also at the coordinate point (X: 10.52 mm, Y: −23.2 mm, Z = 1552.1 mm), on the intrados of the blade (cross-sections, views from the tip of the blade).

Equivalent (von-Mises) Stress	82.62 MPa
Maximum Principal Stress	88.85 MPa
Middle Principal Stress	16.70 MPa
Minimum Principal Stress	−1.28 MPa

## Data Availability

The data presented in this study are available on request from the corresponding author.

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
