# Peer review of "Analysis of the Effect of Fiber Orientation on Mechanical and Elastic Characteristics at Axial Stresses of GFRP Used in Wind Turbine Blades"

_polymers, 2023, doi:10.3390/polym15040861_

Round 1

Reviewer 1 Report

The paper addresses the mechanical behavior of multiple layers of GFRP at different ply orientations. The mechanical properties addressed include tensile and compressive properties. The work cannot be considered as a new contribution but affirmative of existing knowledge and to some extent provide new information about the materials properties. Tests carried are very sensitive and requires high laboratory skills. Before publication the following suggestions are strongly recommended:

The authors should be careful about saying : “The mechanical properties of GFRP composites make these materials ideal for building and strengthening structures.” Indeed, the use of GFRP materials in building (either as a standalone material or as internal/external reinforcement) is still hindered by the concerns associated with their behaviour under elevated temperatures. Please include this aspect in the state of the art by checking this document: 10.1016/j.conbuildmat.2022.128340

At the end of the introduction section, the aim of study should be more emphasized.

Did the authors perform tests on the resin? The authors should provide some information about this. Even the data provided by the manufacturer can be useful.

The amount of fibres positioned in each direction is an important parameters that affect the stiffness and strength properties of composites. Please provide this info in a table.

Could the authors specify the load-cell capacity of the UTM used in the experimental campaign?

To have a better understanding about the influence of the 45° plies, why the authors did not perform tests in shear? If feasible, the reviewer strongly suggests performing these tests

How were strains measured in the compression tests?

The authors should highlight the differences of failure mechanisms in composites made of different fibre layup.

At the end of the results section, the reviewer suggests making a comparison with data available in the literature to spot some general trend about the variation of the mechanical properties with changes in the layup sequence. This can be very useful for other researchers working on a similar topic.

In the conclusions, the authors should explain the significance and shortcomings of the research work, instead of repeating the results obtained before.

Author Response

Response to Reviewer 1 Comments

The paper addresses the mechanical behavior of multiple layers of GFRP at different ply orientations. The mechanical properties addressed include tensile and compressive properties. The work cannot be considered as a new contribution but affirmative of existing knowledge and to some extent provide new information about the materials properties. Tests carried are very sensitive and requires high laboratory skills. Before publication the following suggestions are strongly recommended:

Dear reviewer,

Thank you very much for your positive feedback. You ask to clarify more details on the mechanical behavior of GFRP, which certainly would give an improvement for our research. So, thank you for these valuable comments. We have tried to answer to all aspects that were mentioned.

Point 1: The authors should be careful about saying: “The mechanical properties of GFRP composites make these materials ideal for building and strengthening structures.” Indeed, the use of GFRP materials in building (either as a standalone material or as internal/external reinforcement) is still hindered by the concerns associated with their behaviour under elevated temperatures. Please include this aspect in the state of the art by checking this document: 10.1016/j.conbuildmat.2022.128340.

At the end of the introduction section, the aim of study should be more emphasized.

Response 1:

We have carefully read the paper indicated and noticed how differently GFRP behaves when temperatures are elevated. As a result, in order to inform our readers about the changes that occur in the mechanical and elastic characteristics of materials at elevated temperatures, we have inserted these aspects in the introductory part, with the appropriate citation – lines (103-111).

Point 2: Did the authors perform tests on the resin? The authors should provide some information about this. Even the data provided by the manufacturer can be useful.

Response 2:

In chapter Materials, we have introduced data about the characteristics of the used resin. These data are now found in the newly introduced Table 1.

Point 3: The amount of fibres positioned in each direction is an important parameters that affect the stiffness and strength properties of composites. Please provide this info in a table.

Response3:

In the paper, at the end of Chapter 2, lines 190.-196, we have included the details below, to show clearly how the fibre overlaps have been carried out in practice by the company that supplied them to us. Also shown in Figures 1 and 2 are the layouts and quantity of fibres used in each reinforcement direction.

“Composite plate type [0°/90°] contains 5 pieces of tissue arranged unidirectionally at 0° and another 5 pieces of tissue arranged unidirectionally at 90°. Each piece contains overlapping glass fibers rowing bound together as if we had two layers of unidirectional glass fibers rowing at 0° and 90°. In the end, this results in 10 alternately overlapping unidirectional layers, 5 of which are oriented at 0° the others being oriented at 90°. In the same way the composite plate [-45°/0°/+45°/90°] was made, the sequence of unidirectional layers being as follows: /-45°/+45°/0°/90°/+45°/-45°/90°/0°/-45°/+45°/.”

Point 4: Could the authors specify the load-cell capacity of the UTM used in the experimental campaign?

Response 4:

The tests were carried out on the INSTRON 8801 universal machine that developing a maximum force of 100KN

Point 5: To have a better understanding about the influence of the 45° plies, why the authors did not perform tests in shear? If feasible, the reviewer strongly suggests performing these tests.

Response 5:

We have introduced in the paper in the chapter Results, lines (333-336 )

“In the paper [33] shear tests were presented with the Iosipescu method [ASTM D5379], on specimens taken from the same composite material and τ12= 54MPa was obtained, on specimen reinforced at [0°/90°]. A comparison between the shear test methods for composite materials shows that the Iosipescu method is considered the most accurate [34]”.

Point 6: How were strains measured in the compression tests?

Response 6:

Compression strains has been taken over from the test machine. The machine software works on the basis of the following calculation relation:

ε=Δl/l0 where ε is the compression strain, Δl is the linear compression deformation and l0 is the length between the aluminium plates, being the same area where the test machine grips were clamped.

Point 7: The authors should highlight the differences of failure mechanisms in composites made of different fibre layup.

Response 7:

We have introduced in the paper in the chapter Results, lines (268-274)

“The three directions (TR, LG and diagonal at 45°) of the plate [0°/90°] and two directions (TR and LG) of the plate [-45°/0°/+45°/90°] were analysed for failure type and failure mode. These aspects complied with the stress test failure codes/typical modes of ASTM D3039. In the case of the specimens from the 0-90 plate on the LG and TR direction they comply with the DGM (edge delamination gage middle) tensile test failure codes, but on the 45 direction we find the AGM (2) - angled gage middle code.”

“For the plate with orientation at [-45°/0°/+45°/90°] on the TR direction the code XGM- explosive gage middle was identified and on the LG direction the failure mode was AGM (1)- angled gage middle.” – ( line 299-302)

 In order to visualize all types of failure, a picture from ASTM D3039 was attached.

Point 8: At the end of the results section, the reviewer suggests making a comparison with data available in the literature to spot some general trend about the variation of the mechanical properties with changes in the layup sequence. This can be very useful for other researchers working on a similar topic.

Response 8:

Unfortunately, not all researchers provide the layer storage sequences needed to see similarities and differences in both fibre orientation and quantity used. In the paper [22] where the research was carried out on the same type of GFRP, the number of layers used is mentioned, four and three. The similar characteristics obtained have similar values. However, we cannot make comparisons with only one paper, as for the others we do not have data available under the same conditions of composite plates formation.

Point 9: In the conclusions, the authors should explain the significance and shortcomings of the research work, instead of repeating the results obtained before.

Response 9:

In the final part of the paper, in the conclusion chapter, we have introduced additional information on the motivation and importance of the present research as well as on the future research directions that should be undertaken in order to have a more complete view on the influence of the fibre orientation mode, as well as the fibre volume, on the main mechanical and elastic characteristics of GFRP used in wind turbine blades.

We hope that our answers meet your requirements.

Reviewer 2 Report

The manuscript having ID: polymers-2190541 is focused on the effects of fiber orientation in wind turbine blade and corresponding mechanical performance. Samples in different directions were cut from a single fabricated glass composite structure. Tensile testing, compression tests, and SEM were performed in addition to the FEA study.

The work presented in this study is lacking novelty.  I do not recommend this article. Following are my comments:

1.                         What is novelty. It is completely hidden and have not been highlighted in the manuscript.   

2.                         The language of the abstract needs significant improvement.

3.                         Introduction is not very focused. It needs improvement and a good discussion relevant to the study must be added.

4.                         Zoom in Figure 1-a and remove extra background. Also enhance the font size there-in.

5.                         Overall, graphs in figures needs enhancement. Pick any article and follow to make good quality plots.

Author Response

Response to Reviewer 2 Comments

The manuscript having ID: polymers-2190541 is focused on the effects of fiber orientation in wind turbine blade and corresponding mechanical performance. Samples in different directions were cut from a single fabricated glass composite structure. Tensile testing, compression tests, and SEM were performed in addition to the FEA study.

 The work presented in this study is lacking novelty.  I do not recommend this article. Following are my comments:

Dear reviewer,

Thank you very much for your feedback. You ask to clarify more details on the effects of fiber orientation in wind turbine blade, which certainly would give an improvement for our research. So, thank you for these valuable comments. We have tried to answer to all aspects that were mentioned.

Point 1: What is novelty. It is completely hidden and have not been highlighted in the manuscript.

Response 1:

In the literature, not much data is presented regarding the determination of the Poisson's ratio on three different directions of a composite material. From our point of view, the ratio between the elongation in one direction and the shrinkage in the direction at 90° is very important in the operation of wind turbine blades. In the paper, we have performed this on two types of plates with different fiber orientation. In the paper we used the Tsai-Hill criterion which is an important element in the calculation of the tensile strength, using data obtained experimentally from tests on the material used.

On the other hand, both at the end of the introductory part and in the conclusions (see bellow), we have highlighted both the novelty and contributions of the authors and the directions for future research to complete this work.

“Based on the tests carried out, among the reinforcement solutions adopted, it was determined which is the best solution for use in wind turbine blades. It is taken into account that, during the loading, the direction of the maximum stresses may change, thus unfavorably orienting the fibres. The novelty of this work lies in the fact that both mechanical and elastic characteristics were determined in three different directions with respect to the fibre orientation directions, using the experimental values obtained in the finite element analysis. SEM analyses of the fracture surfaces revealed the character and mode of fracture under tensile stresses.” Line-(150-157)

Consequently, the study carried out in this paper shows that the arrangement of the fibres in relation to the direction of the maximum stress is very important. The stresses directions in operation are relatively random and the direction of the maximum stress can differ substantially, for example in relation to the wind direction. If we consider the material [0°/90°], if the direction of maximum stress were at 45°, there would be a rapid deterioration of the wind turbine blades due to both a decrease in tensile characteristics and a substantial change in Poisson's ratio with respect to the other two directions, 0° and 90°. Particular attention should be paid to the orientation of the fibres especially in the areas of maximum stress revealed by the finite element analysis. In this work, tests were also carried out on the [-45°/0°/+45°/90°], plate, from which significant improvements were found in terms of the much smaller variation of the characteristics in relation to the direction of fibre orientation. Here again it should be taken into account that 10 layers can still be used, leaving fewer layers with orientation at 900. Considering that the weight of the wind turbine blade should be as low as possible, adding more fibres is not an option. As a result, at a minimum volume of fibres used, maximum strength should be achieved, and this is achieved through proper fibre orientation.

Future research should focus on:

- Optimising the volume of fibres added in relation to their orientation;

- Determinations similar to those carried out in this paper on other types of fibre arrangements;

- Measurements on the turbine blade at full or reduced scale to determine the variation of the direction of the maximum normal and shear stresses in relation to different ratios of tensile and torsional loading.” Line- (555-575)

Point 2: The language of the abstract needs significant improvement

Response 2:

The language of the abstract has been significantly improved.

 Due to its physical and mechanical properties, glass fiber-reinforced polymer (GFRP) is utilised in wind turbine blades. The loads given to the blades of wind turbines, particularly those operating offshore, are relatively significant. In addition to the typical static stresses, there are also large dy-namic stresses, which are mostly induced by wind direction changes. When the maximum stresses resulting from fatigue loading change direction, the reinforcing directions of the material used to manufacture the wind turbine blades must also be considered. In this study, sandwich-reinforced GFRP materials were subjected to tensile testing in three directions. The parameters of the stress-strain curve were identified based on the three orientations in which samples were cut from the original plate. Strain gauge sensors were utilized in order to establish the three-dimensional elasticity of a material. After a fracture was created by tensile stress, SEM images were taken to highlight the fracture's characteristics. Using finite element analyses, the stress strain directions were determined. In accordance to the three orientations and the various reinforcements used, it was established that the wind turbine blades are operational.”

Point 3: Introduction is not very focused. It needs improvement and a good discussion relevant to the study must be added.

Response3:

Within the chapter on the introduction, we have inserted two paragraphs on:

- Influence of temperature on material characteristics (requested by one of the reviewers);

- News and importance of the research carried out in the paper;

- Arguments regarding the research methods carried out in the paper.

All these paragraphs are now highlighted in yellow.

Point 4: Zoom in Figure 1-a and remove extra background. Also enhance the font size there-in.

Response 4:

The additional background has been removed as well as the dimensions have been changed, for figures 1 and 2.

Point 5: Overall, graphs in figures needs enhancement. Pick any article and follow to make good quality plots.

Response 5:

All graphs in the figures of the paper have been considerably improved using Origin 2019 software, (figures 6,8,10,11,12)

We hope that our answers meet your requirements.

Round 2

Reviewer 1 Report

The paper is significantly improved following the review process. The authors responded to all the reviewers' comments, therefore, I recommend that the paper be published in the current form. Of course, the final decision belongs to the Editor.

Author Response

Dear reviewer,

Thank you very much for your positive feedback and wea are very happy that you accepted the current form.

Best regards,

Reviewer 2 Report

Dear Authors,

The manuscript has been improved and the novelty has been highlighted in the revised submission. Respond to the following comment:

1. On page 497, Why Tsai-Hill criterion was used for prediction of damage in FRPCs. Include justification of why it was used? Include the following article in references.

https://doi.org/10.1016/j.compstruct.2018.06.010

Author Response

Response to Reviewer 2 Comments

“The manuscript has been improved and the novelty has been highlighted in the revised submission. Respond to the following comment”:

Dear reviewer,

Thank you very much for your feedback. We have tried to answer to all aspects that were mentioned.

 Point 1: On page 497, Why Tsai-Hill criterion was used for prediction of damage in FRPCs. Include justification of why it was used? Include the following article in references. https://doi.org/10.1016/j.compstruct.2018.06.010

Response 1:

The Tsai-Hill yield criterion is widely used for anisotropic composite materials that have different tensile and compressive strengths, in this case the GFRP used in WTB construction. The Tsai-Hill criterion predicts failure when the failure index in a laminate reaches a value of 1. I have carefully read the paper indicated and noticed how differently GFRP behaves when the Puck criterion is used and how the fibers yield. We have inserted these aspects in the results part, with the appropriate citation.

“In the paper [42] a new elastoplastic continuum damage model, based on Puck’s theory, is presented. The model is intended for plane state of stress. It is capable to describe the non-linearity due to the irreversible strain in the matrix, but cannot be used for composites with the textile plies. Taking into account the particularities of the tested material, the Tsai-Hill criterion will be used to calculate the WTB analyzed above.”. – lines (496-500).
